# First-Principles Studies for Electronic Structure and Optical Properties of Strontium Doped β-Ga_2_O_3_

**DOI:** 10.3390/mi12040348

**Published:** 2021-03-24

**Authors:** Loh Kean Ping, Mohd Ambri Mohamed, Abhay Kumar Mondal, Mohamad Fariz Mohamad Taib, Mohd Hazrie Samat, Dilla Duryha Berhanuddin, P. Susthitha Menon, Raihana Bahru

**Affiliations:** 1Institute of Microengineering and Nanoelectronics (IMEN), Universiti Kebangsaan Malaysia (UKM), Bangi 43600, Selangor, Malaysia; p103776@siswa.ukm.edu.my (L.K.P.); abhay.nano17@gmail.com (A.K.M.); dduryha@ukm.edu.my (D.D.B.); susi@ukm.edu.my (P.S.M.); raihanabahru@ukm.edu.my (R.B.); 2Faculty of Applied Sciences, Universiti Teknologi MARA (UiTM), Shah Alam 40450, Selangor, Malaysia; mfariz@uitm.edu.my (M.F.M.T.); mohdhazrie@uitm.edu.my (M.H.S.); 3Ionic Materials & Devices (iMADE) Research Laboratory, Institute of Science, Universiti Teknologi MARA (UiTM), Shah Alam 40450, Selangor, Malaysia

**Keywords:** first-principles, density functional theory, pure β-Ga_2_O_3_, Sr-doped β-Ga_2_O_3_, p-type doping, band structure, density of states, optical absorption

## Abstract

The crystal structure, electron charge density, band structure, density of states, and optical properties of pure and strontium (Sr)-doped β-Ga_2_O_3_ were studied using the first-principles calculation based on the density functional theory (DFT) within the generalized-gradient approximation (GGA) with the Perdew–Burke–Ernzerhof (PBE). The reason for choosing strontium as a dopant is due to its p-type doping behavior, which is expected to boost the material’s electrical and optical properties and maximize the devices’ efficiency. The structural parameter for pure β-Ga_2_O_3_ crystal structure is in the monoclinic space group (C2/m), which shows good agreement with the previous studies from experimental work. Bandgap energy from both pure and Sr-doped β-Ga_2_O_3_ is lower than the experimental bandgap value due to the limitation of DFT, which will ignore the calculation of exchange-correlation potential. To counterbalance the current incompatibilities, the better way to complete the theoretical calculations is to refine the theoretical predictions using the scissor operator’s working principle, according to literature published in the past and present. Therefore, the scissor operator was used to overcome the limitation of DFT. The density of states (DOS) shows the hybridization state of Ga 3d, O 2p, and Sr 5s orbital. The bonding population analysis exhibits the bonding characteristics for both pure and Sr-doped β-Ga_2_O_3_. The calculated optical properties for the absorption coefficient in Sr doping causes red-shift of the absorption spectrum, thus, strengthening visible light absorption. The reflectivity, refractive index, dielectric function, and loss function were obtained to understand further this novel work on Sr-doped β-Ga_2_O_3_ from the first-principles calculation.

## 1. Introduction

Gallium oxide (Ga_2_O_3_) embraces five different kinds of polymorphism, such as α, β, γ, δ, and ε phase [1,2]. Other examples of metal oxide structures are lead oxide (Pb_2_O_3_) [3], molybdenum dioxide (MoO_2_) [4], aluminium oxide (Al_2_O_3_) [5], and zirconium oxide (ZrO_2_) [6], which have a variety of polymorph phase similar to Ga_2_O_3_. Among all of these polymorphs of gallium oxide, β-Ga_2_O_3_ plays an essential role in ultrawide bandgap (UWBG) applications, with a bandgap energy of 4.8 eV between the valence band and conduction band [7,8]. This UWBG material has excellent heat thermal stability utilized in power electronics applications [9,10]. Perhaps, history of the monoclinic β-Ga_2_O_3_ can be traced back in the recent past few years due to its stable properties that eventually draw scientists’ profound interest [11,12,13].

One of the significant applications of β-Ga_2_O_3_ which is still being used now is its meta-stable state crystal structure compared to other Ga_2_O_3_ polymorphs [14]. This is because β-Ga_2_O_3_ is a wide bandgap semiconductor with a bandgap (*E*_g_) of about 4.8 eV [15]. The wide-bandgap material can sustain high incoming voltage resulting from large electrical breakdown strength when it comes to short circuit situation. Recently, β-Ga_2_O_3_ has attracted much attention for its potential use in the next-generation optoelectronic devices in the near UV wavelength region. It shows the shortest cut-off wavelength at around 270 nm, whereas conventional transparent conductive oxides (TCOs) belongs to the visible wavelength region. The applications for a monoclinic β-Ga_2_O_3_ structure are such as in solar energy devices [16], passivation coating [17], optoelectronic devices [18], gas sensor [19], and deep ultraviolet radiation devices [20].

The strontium element has the same chemical properties as alkaline earth metals such as beryllium, magnesium, calcium, and barium from group II elements. It is generally used for fireworks and flares purposes [21]. It is also used to produce ferrite magnets and is responsible for zinc’s refining process [22]. One famous application of the strontium-90 radio-isotope is in the military field, especially nuclear weapons [23] and nuclear reactors [24]. The radio-isotope of the strontium-90 is the by-product of a nuclear explosion that initially comes from the uranium element’s nuclear fission [25]. It has a half-life of 29 years [26], which was considered a long duration to degenerate its radioactivity. As for the application of semiconductors, strontium ions can be reused for the color television cathode ray tube (CRTs) to avoid X-ray emission [26,27].

There are no reports on the strontium (Sr^2+^) doping, to date with the pure β-Ga_2_O_3_ based on density functional theory (DFT) in the present first-principles study. The importance of this paper is to determine the effect of Sr doping in Ga_2_O_3_ as p-type doping based on its material properties in the simulation structure. Therefore, this paper will focus on the theoretical investigation of the electronic band structure, total and partial density of states, and optical properties of pure and Sr-doped β-Ga_2_O_3_. Besides, this simulation process allow researchers to discover more about the material’s theoretical part.

## 2. Materials and Methods

The calculations were carried out using Cambridge Serial Total Energy Package (CASTEP) code. This simulation can be traced back to the application of density functional theory (DFT), which utilizes the total-energy plane-wave pseudopotential method [28,29]. The exchange-correlation potential effects were handled by the generalized gradient approximation (GGA) with the Perdew–Burke–Ernzerhof (PBE) functional [30,31,32]. Theoretically, the DFT is based on the ground state, which causes the exchange-correlation potential between the excited electrons to be underestimated [33]. Thus, the calculations from GGA-PBE results in lower energy levels above the valence band than that of experimental results. The utilization of the GGA-PBE in this simulation work was used to compare the parameters, cell volume, and cell angle. β-Ga_2_O_3_ crystal structure belongs to the C2/m group, which is monoclinic. Figure 1 shows the crystal structure of the unit cell (1 × 1 × 1) of pure β-Ga_2_O_3_ and Sr-doped β-Ga_2_O_3_, while Figure 2 show the crystal structure of supercell (1 × 2 × 2) of pure β-Ga_2_O_3_ and Sr-doped β-Ga_2_O_3_. The unit cell crystal structure consists of 20 atoms (8 Ga atoms and 12 O atoms), and the supercell crystal structure consists of 80 atoms (32 Ga atoms and 48 O atoms). Ga atoms occupy 4i Wyckoff position at Ga1(0.09050, 0, 0.79460) and Ga2(0.15866, 0.5, 0.31402), whereas the O atoms occupy 4i Wyckoff positions defined by O1(0.1645, 0, 0.1098), O2(0.1733, 0, 0.5632), and O3(−0.0041, 0.5, 0.2566) [34]. For obtaining exact band gaps and optical properties, the scissors operator has been carried out.

The electronic interaction between the valence electrons and conduction holes was modeled using ultra-fine quality pseudopotentials with an energy cut-off of 380 eV. This cut-off energy was able to bring out the optimized results of the band structure, density of states, electron density, and optical properties calculations. As for the Monhorst Pack scheme *k*-point grid sampling of the reduced Brillouin zone, 1 × 1 × 1 and 1 × 2 × 2 *k*-points were set for pure β-Ga_2_O_3_ and Sr-doped β-Ga_2_O_3,_ respectively. The valence electronic configurations for Ga, O, and Sr are 3d^10^4s^2^4p^1^, 2s^2^2p^4^, and 3d^10^4p^6^5s^2^, respectively. During the geometry optimization, the cut-off energy for both the structure model was 380 eV, the energy convergence for this structure is 1.209 × 10^−6^ eV/atom, the maximum displacement is 5 × 10^−4^ Å, maximum stress is 0.02 GPa, and maximum force is 0.01 eV/Å.

## 3. Results and Discussion

### 3.1. Structural Properties

The structure of β-Ga_2_O_3_ and Sr-doped β-Ga_2_O_3_ were investigated, and the geometry optimized crystal structure is shown in Figure 1 and Figure 2. Table 1 shows the list of the optimized lattice parameters, cell volume, and angle from GGA-PBE functional. The theoretical value for β-Ga_2_O_3_ from GGA-PBE calculation has a small difference in lattice parameter, cell volume, and angle value compared to the experimental value. The lattice parameter difference is less than 2.29% while for the cell volume, it gives a 6.14% error. As for the Sr-doped β-Ga_2_O_3_ part, the error percentage indicates the differences in theoretical results between β-Ga_2_O_3_ and Sr-doped β-Ga_2_O_3_. The lattice parameter and volume of Ga_2_O_3_ increase after Sr doping. This is similar to the other reports using Mg-doped Ga_2_O_3_. The ionic radius of Mg^2+^ of 0.72 Å is larger than that of Ga^3+^ of 0.62 Å. Thus, it is reasonable that the lattice parameters of Ga_2_O_3_ increases after Mg doping [35]. Furthermore, Zn-doped Ga_2_O_3_ also shows the increase of structural parameter after Zn doping because of the ionic radius of Zn^2+^ of 0.74 Å is larger than that of Ga^3+^ of 0.62 Å, which resulted in the lattice spacing gradually being enlarged [36]. However, to the best of our knowledge, there is no theoretical and experimental data available for Sr-doped β-Ga_2_O_3_ for comparison with this work.

As for Table 2, different doping sites were taken at Ga1 and Ga2 for 1 × 2 × 2 supercells. Sr-doped β-Ga_2_O_3_ has a greater lattice parameter and cell volume compared to pure β-Ga_2_O_3_. This also means that Sr-dopant enlarged the original size of pure β-Ga_2_O_3_. The highest error of estimation for the lattice parameter between Sr-doped β-Ga_2_O_3_ at Ga1 and Ga2 are about 1.11 % indicates that the results are nearly close to each other. 

The average bond length for pure and Sr-doped β-Ga_2_O_3_ is shown in Table 3, while the average bond length for pure and Sr-doped β-Ga_2_O_3_ at Ga1 and Ga2 in 1 × 2 × 2 supercell were listed in Table 4. It is shown that the atomic radius of the Ga atom (1.36 Å) is smaller than the Sr (2.19 Å), according to the periodic table. Therefore, Ga-O bond length is shorter than Sr-O and O-O bonds in Sr-doped β-Ga_2_O_3_. This situation is the same as the pure β-Ga_2_O_3,_ where Ga-O bond length is shorter than O-O bonds. After Sr doping, the overall bond length for Sr-doped β-Ga_2_O_3_ is increased more than pure β-Ga_2_O_3_ due to the presence of Sr^2+^ ions.

The spatial electron density maps determine whether the structure model belongs to ionic or covalent bonds. Figure 3 exhibits the distribution of the different structure’s electron density in pure and Sr-doped β-Ga_2_O_3_. It shows that pure β-Ga_2_O_3_ has strong covalent bonding characteristics before doping. On the other hand, the bond population analysis for Sr-doped β-Ga_2_O_3_ exhibits a weak ionic bonding effect compared to pure β-Ga_2_O_3_. This is because, the Sr^2+^ dopant had changed the bonding characteristics of β-Ga_2_O_3_ after doping. Table 5 shows the bond population analysis indicator for electron density distribution in pure and Sr-doped β-Ga_2_O_3_. It can be determined from Figure 3a that pure β-Ga_2_O_3_ shows strong covalent bonding characteristics. This is because the Ga and O atoms were located at the strong covalent region nearby to the red color region. On the other hand, there were weak ionic bonding characteristics in Sr-doped β-Ga_2_O_3,_ where some of the Ga and O atoms are located in the green color indicator region.

### 3.2. Properties of Electronic Structure

This section will discuss the electronic structure properties used to determine the band structure, total and partial density of states (DOS) of β-Ga_2_O_3_. Gallium oxide displays an indirect bandgap of 4.8–4.9 eV, both for the experimental and simulation work [38,39]. Besides, the application of Ga_2_O_3_ is applied in power MOSFETs, where high breakdown electric field and large Balinga’s figure of merit occur [40]. The (100) plane of Ga_2_O_3_ is mostly taken in experimental work because it gives a high resolution of result analysis which takes places at the specific Brillouin zone, Γ-Z and A-M directions [41]. This unique electronic structure properties of Ga_2_O_3_ bring out new hope in future applications, especially in electronics, optoelectronics, and sensing systems [7].

The bandgap is usually measured between the conduction band minimum (CBM) and the valence band maximum (VBM), located at the Fermi level. The high-symmetry direction of the Brillouin zone of pure β-Ga_2_O_3_ along with the G-F-Q-Z-G path, is illustrated in Figure 4. Figure 5a shows that pure β-Ga_2_O_3_ has a bandgap energy of 1.939 eV, lower than the bandgap energy from the experimental work but consistent with other calculated results from DFT [42,43]. The measured bandgap energy for experimental work is 4.8 eV [15]. This phenomenon can be explained by the underestimation of density functional theory (DFT) limitations. Therefore, the calculations of band structures with scissor operator were considered to overcome bandgap underestimation from the DFT method. The scissor operator was introduced to shift all the conduction levels to agree with the band gap’s measured value [44]. In our case, the scissor operator’s value for unit cell 1 × 1 × 1 was taken to be 4.8 − 1.939 = 2.861 eV, accounting for the difference between the experimental band gap (4.8 eV) [15] and the calculated GGA bandgap (1.939 eV) for β-Ga_2_O_3_. For supercell 1 × 2 × 2, the scissor operator’s value is 2.868 eV. The band structure of pure β-Ga_2_O_3_ in P1 symmetry along with G-F-Q-Z-Q path is shown in Figure 5 to compare with the band structure of Sr-doped β-Ga_2_O_3_ also in P1 symmetry. Figure 5b shows that the bandgap energy decreases to 1.879 eV after Sr doping. This indicates that Sr^2+^ ions possess p-type doping behavior, which creates more holes to accept electrons that allow the semiconductor to perform efficiently during the presence of conducting current. Figure 6 presents the band structures of pure β-Ga_2_O_3_ and Sr-doped β-Ga_2_O_3_ with a scissor operator. It shows that the bandgap of pure β-Ga_2_O_3_ was corrected to match the experimental band gap at 4.8 eV [15] while the bandgap of Sr-doped β-Ga_2_O_3_ is 4.740 eV which is lower than pure β-Ga_2_O_3_.

As for Figure 7, it is the band structure for 1 × 2 × 2 supercell for pure and Sr-doped β-Ga_2_O_3_. Pure β-Ga_2_O_3_ has a bandgap of 1.932 eV. On the other hand, the Sr-doped at different doping sites at Ga1 and Ga2 have bandgaps of 1.826 and 1.840 eV, respectively. The band structure for unit cell and supercell show slight differences in bandgap, according to this investigation. For band structures with scissor operator of 1 × 2 × 2 supercell for pure and Sr-doped β-Ga_2_O_3_ in Figure 8, the bandgap of pure β-Ga_2_O_3_ in 1 × 2 × 2 supercell is 4.8 eV, and the bandgap of Sr-doped β-Ga_2_O_3_ at Ga1 and Ga2 is 4.694 and 4.708 eV, respectively.

Figure 9 shows the total and partial DOS of pure and Sr-doped β-Ga_2_O_3_. Pure β-Ga_2_O_3_ comprises of Ga 4s at the top of the valence band and O 2p located at the bottom of the conduction band. As for the Sr-doped β-Ga_2_O_3_, the Sr atom has an atom of s orbital and introduces dopant energy levels in the pure β-Ga_2_O_3_. The valence band mainly consist of Sr 5s, Sr 4p, O 2p, and Ga 3d at −32, −12.5, −2, and −13 eV, respectively. The conduction band was dominated by Sr 3d, O 2p, and Ga 4p. Therefore, Sr dopant changes the covalent bonding characteristics of pure β-Ga_2_O_3_. This also indicates that weak ionic bonding characteristic appears between Ga, O, and Sr atoms. Figure 10 shows the total and partial DOS for 1 × 2 × 2 supercell of pure and Sr-doped β-Ga_2_O_3_ at different doping sites Ga1 and Ga2. The same hybridization states occur in the valence band, such as Sr 5s, Sr 4p, O 2p, and Ga 3d. The conduction band is also dominated by the same hybridization states of Sr 3d, O 2p, and Ga 4p. All these results could be determined when comparing 1 × 1 × 1 unit cell and 1 × 2 × 2 supercell.

### 3.3. Optical Properties

Optical properties’ importance is usually highlighted in the absorption coefficient, reflectivity, refractive index, dielectric function, and loss function. All these optical properties features are related to the complex dielectric function formula, which is written as in Equation (1):(1)$$\varepsilon \left(\omega \right)=\varepsilon_{1}\left(\omega \right)+i\varepsilon_{2}\left(\omega \right).$$

The ***ε*_1_(*ω*)** and ***ε_2_(ω)*** are the real and imaginary part, respectively. The real part is correlated to the degree of electronic polarization and calculated from the Kramers–Kronig relation. On the other hand, the imaginary part is associated with the material’s dielectric losses. All other optical properties can be derived from $\varepsilon_{1}\left(\omega \right)$ and $\varepsilon_{2}\left(\omega \right)$ by the Kramer–Kronig relation.

The other well-known formula for optical properties such as absorption (**α**(**ω**)), reflectivity (***R***(***ω***)), refractive index (***n***(***ω***)), and loss function (***L***(***ω***)) is defined as follows in Equations (2)–(5):(2)$$\alpha \left(\omega \right)=\frac{4\pi k}{\lambda }=\frac{2k\omega }{c}=\sqrt{2}\omega {\left[\sqrt{\varepsilon_{1}^{2}\left(\omega \right)+\varepsilon_{2}^{2}\left(\omega \right)}-\varepsilon_{1}\left(\omega \right)\;\right]}^{1/2},$$
(3)$$R\left(\omega \right)={\left|\frac{1-\;\sqrt{\varepsilon \left(\omega \right)}}{1+\varepsilon \left(\omega \right)}\right|}^{2},$$
(4)$$n\left(\omega \right)=\sqrt{\frac{\left|\varepsilon \left(\omega \right)\right|+\varepsilon_{1}\left(\omega \right)}{2}},$$
(5)$$L\left(\omega \right)=Im\left[\frac{-1}{\varepsilon \left(\omega \right)}\right]=\frac{\varepsilon_{2}\left(\omega \right)}{\varepsilon_{1}^{2}\left(\omega \right)+\varepsilon_{2}^{2}\left(\omega \right)}.$$

The band structure of pure β-Ga_2_O_3_ shows an underestimated bandgap value of 1.939 eV compared to the experimental bandgap. Such underestimation of calculated bandgap values is a common feature of the DFT calculations and can be overcome by applying the so-called scissor operator [45]. Such a correction is significant for calculations of the optical properties. To facilitate comparison with the experimental results, we utilized a scissor operator to match the calculated optical gap determined via experimental techniques. The calculated optical properties of pure Ga_2_O_3_ and Sr-doped β-Ga_2_O_3_ without and with scissor operator are presented in Table 6 and Table 7. For the absorption coefficient, the absorption edge with the scissor operator shifts the light absorption towards the UV light region, corresponding to the bandgap value of β-Ga_2_O_3._ This result is consistent with other experimental results for absorption spectra in a deep UV-Vis range of 200 to 300 nm using UV-Vis spectroscopy [46]. The scissor operator’s shifts the major peak of reflectivity and loss function towards higher photon energy. For the refractive index of pure Ga_2_O_3_ and Sr-doped β-Ga_2_O_3_, the scissor operator decreases its value by 11.4% and 31.8%, while the dielectric constant decreases by 21.6% and 52.9% as compared to the calculations without scissor operator. Figure 11, Figure 12 and Figure 13 present the optical properties using a scissor operator.

Figure 11 presents the absorption spectrum of pure and Sr-doped β-Ga_2_O_3_ structure in 1 × 1 × 1 unit cell and 1 × 2 × 2 supercell. It can be observed that the absorption region is a broad spectrum. The most important wavelength emission for pure and Sr-doped β-Ga_2_O_3_ system mainly resides in the deep UV region, as shown in Figure 11d. After Sr-doping, the wavelength emission increases as the bandgap energy of Sr-doped β-Ga_2_O_3_ decreases. This simulation result matches with the principle of the Einstein–Planck relation: E=hf=hc/λ, where its bandgap energy decreases and increases the wavelength emission spectrum.

Meanwhile, this also indicates that the emission spectrum has been red-shifted after Sr-doping, proving that Sr dopant possesses p-type doping behavior. There is no available absorption spectrum report for Sr-doped β-Ga_2_O_3_ for comparison with this theoretical work. However, this finding is similar to other dopants in Ga_2_O_3,_ such as Mg-doped Ga_2_O_3_ [46] and Zn-doped Ga_2_O_3_ from experimental work, which decreases the bandgap after doping and also possesses p-type doping behavior. The results for Mg-doped Ga_2_O_3_ suggested that it is a promising material candidate for solar-blind photodetector due to its lower dark current, higher sensitivity, and faster decay time which can be attributed to the high insulating and low defect concentration. For Zn-doped Ga_2_O_3_, its bandgap is 4.90–4.93 eV for different Zn doping contents which is reduced by 0.20–0.81% compared to pure β-Ga_2_O_3_ (4.94 eV). This is agreeable with our theoretical work, which decreases the bandgap after Sr doping by 1.25–5.49%.

Figure 12 and Figure 13 shows the reflectivity, refractive index, dielectric function, and loss function of pure and Sr-doped β-Ga_2_O_3_ for both the 1 × 1 × 1 unit cell and 1 × 2 × 2 supercell. The energy range is shown as ~0–25 eV in this simulation work. For the energy spectrum of reflectivity, the energy increases slightly to 16 eV. Simultaneously, this incident causes a decrease of reflectivity after the Sr-doping, which eventually affects surface material effectiveness to reflect the electromagnetic radiation energy. More input electromagnetic radiation energy cannot be reflected completely and finally reside in the Sr-doped β-Ga_2_O_3_ structure compared to pure β-Ga_2_O_3_.

The refractive index exhibits a different growing emission spectrum trend compared to reflectivity. It has a real (*n*) and imaginary (*k*) part for both pure and Sr-doped β-Ga_2_O_3_. The emission spectrum releases energy in the range of ~0–25 eV for both systems. However, there are still differences in its refractive index spectrum in both real (*n*) and the imaginary (*k*) parts between pure and Sr-doped β-Ga_2_O_3_. The refractive index of Sr-doped β-Ga_2_O_3_ is 1.59, which is slightly higher than pure β-Ga_2_O_3_ (1.55).

As for the dielectric function, this feature of optical properties is usually observed in the spectrum’s imaginary part (ε_2_). It has dielectric constant, ε_0_ = 2.39 for pure Ga_2_O_3_ and ε_0_ = 2.53 for Sr-doped Ga_2_O_3_. The imaginary part’s major peaks are located at 11.5 and 11.8 eV for pure and Sr-doped β-Ga_2_O_3_. This result indicates that the Sr dopant could affect the optical properties to determine the energy range. After the Sr-doping, the peak energy spectrum increases compared to pure β-Ga_2_O_3_. The peak energy shift also means a shift in the localized degree of a free electron and holes between the conduction and valence band in the impurity doping structure.

Lastly, the loss function feature in the optical properties is often used to determine the energy loss of the free electrons crossing along with the material. The major peak energy for the Sr-doped β-Ga_2_O_3_ is located at 16.3 eV, and the pure β-Ga_2_O_3_ is located at 17.4 eV, which is lower than Sr-doped β-Ga_2_O_3_. The Sr doping causes a decrease in the energy loss function of β-Ga_2_O_3_. This indicates that the Sr dopant tends to reduce its energy loss and improve its emission of peak energy efficiency for better performance in the material, matching with the characteristic of p-type doping material. It can withstand high voltage and current with minimum energy loss.

## 4. Conclusions

In conclusion, the first-principles studies provided calculations of the structural, electronic, and optical properties of pure and Sr-doped β-Ga_2_O_3_. There are not many differences in the calculated lattice parameter, cell volume, and angle of the pure β-Ga_2_O_3_ structure at GGA-PBE with experimental data. GGA-PBE is still preferable for the calculated structural parameter as it has good theoretical results, which almost matches the experimental work. Pure β-Ga_2_O_3_ has an indirect bandgap of 1.939 eV while for Sr-doped β-Ga_2_O_3_, the bandgap is 1.879 eV for 1 × 1 × 1 unit cell. Meanwhile, the 1 × 2 × 2 supercell for pure β-Ga_2_O_3_ has a bandgap of 1.932 eV and Sr-doped β-Ga_2_O_3_ at different doping sites Ga1 and Ga2 have bandgaps of 1.826 and 1.840 eV, respectively. This bandgap was underestimated compared to the experimental value, so the scissor operator was used to correct the bandgap. The decrease in bandgap energy was due to the creation of more holes to accept more incoming free electrons, indicating a p-type doping behavior for Sr dopant. The electronic interaction between the valence band and conduction band contains O 2p and Ga 4s orbital for pure β-Ga_2_O_3_. On the other hand, Sr-doped β-Ga_2_O_3_ consists of Sr 5s, Sr 4p, O 2p, and Ga 3d for the electronic interaction between valence and conduction band. These hybridization states can be observed from the total and partial DOS for both 1 × 1 × 1 unit cell and 1 × 2 × 2 supercell. The population analysis for pure β-Ga_2_O_3_ was considered as strong covalent bonding characteristics. As for the Sr-doped β-Ga_2_O_3_, it changes its bonding characteristics to weak ionic bonds. The optical absorption for Sr-doped β-Ga_2_O_3_ exhibited a red-shifted spectrum compared to pure β-Ga_2_O_3_. This matches the optical behavior for p-type doping, where the material emits broader wavelength emission, and a red-shifted spectrum occurred. The optical absorption for both pure and Sr-doped β-Ga_2_O_3_ was found in the deep ultraviolet light (DUV) region according to the absorption coefficient.

## Figures and Tables

**Figure 1 micromachines-12-00348-f001:**
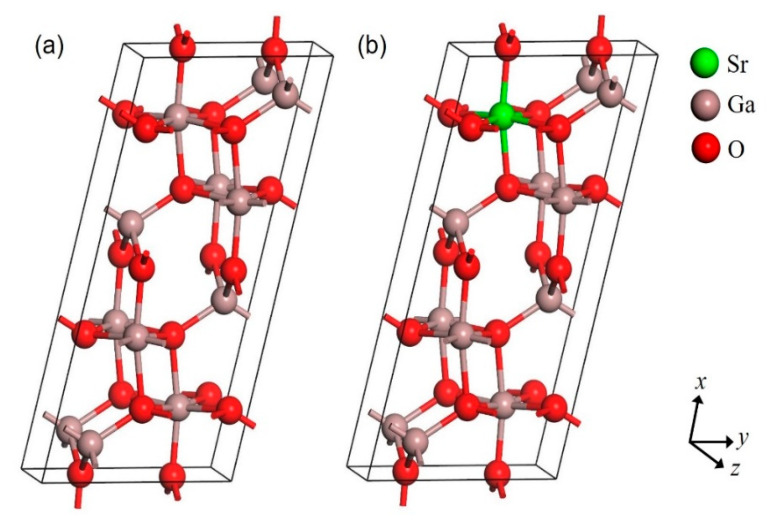
Crystal structure of 1 × 1 × 1 unit cell of (**a**) pure β-Ga_2_O_3_ and (**b**) Sr-doped β-Ga_2_O_3._

**Figure 2 micromachines-12-00348-f002:**
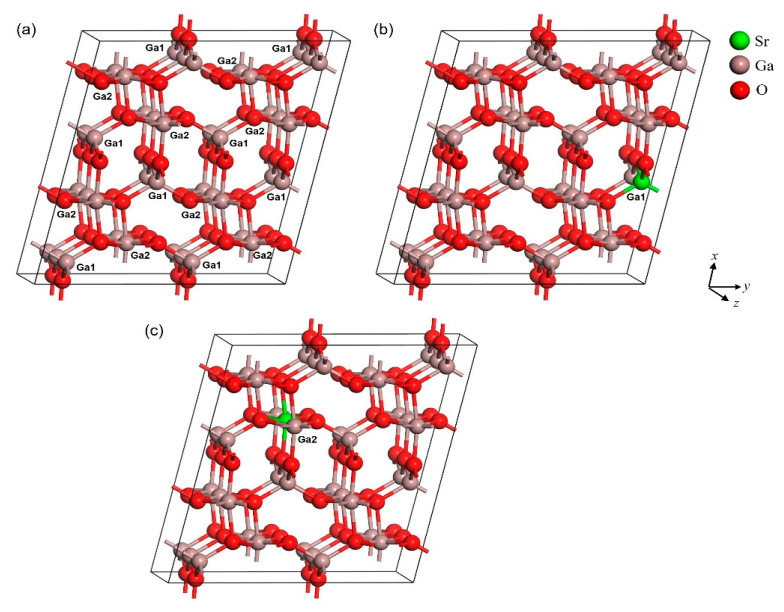
Crystal structure of 1 × 2 × 2 supercell of (**a**) pure β-Ga_2_O_3_, (**b**) Sr-doped β-Ga_2_O_3_ at Ga1, and (**c**) Sr-doped Ga_2_O_3_ at Ga2.

**Figure 3 micromachines-12-00348-f003:**
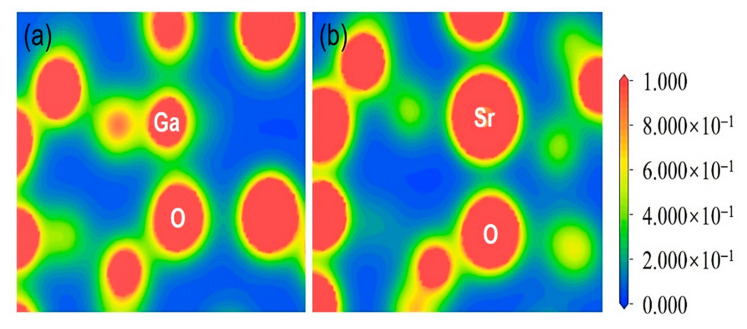
Distribution of electron density of (**a**) pure β-Ga_2_O_3_ and (**b**) Sr-doped β-Ga_2_O_3_.

**Figure 4 micromachines-12-00348-f004:**
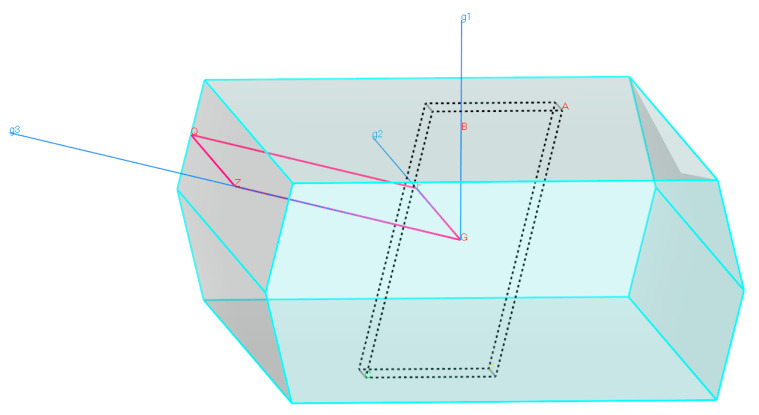
Brillouin zone path at G-F-G-Z-Q direction for β-Ga_2_O_3_ in P1 symmetry.

**Figure 5 micromachines-12-00348-f005:**
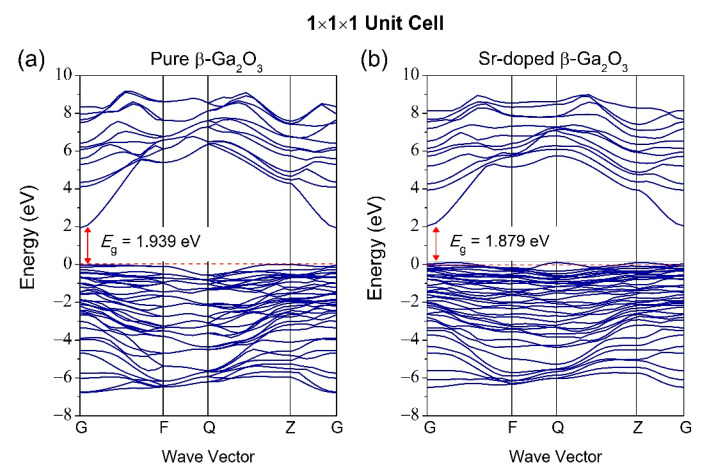
Band structure of (**a**) pure β-Ga_2_O_3_ and (**b**) Sr-doped β-Ga_2_O_3_ without scissor operator.

**Figure 6 micromachines-12-00348-f006:**
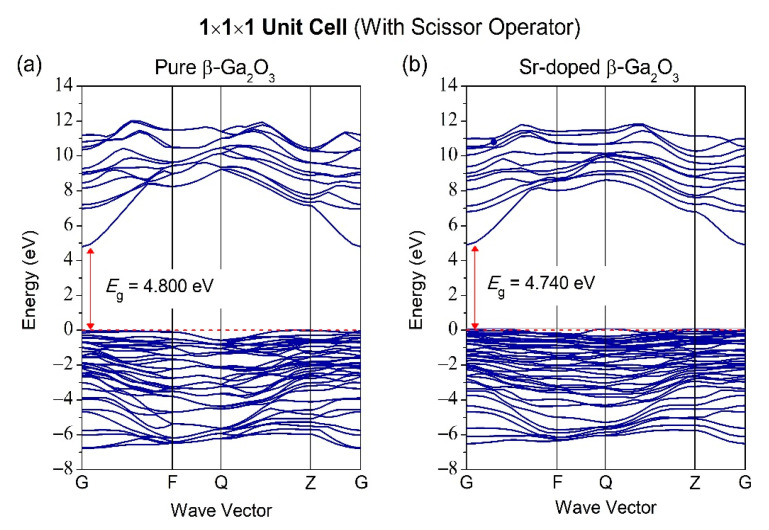
Band structure of (**a**) pure β-Ga_2_O_3_ and (**b**) Sr-doped β-Ga_2_O_3_ with scissor operator.

**Figure 7 micromachines-12-00348-f007:**
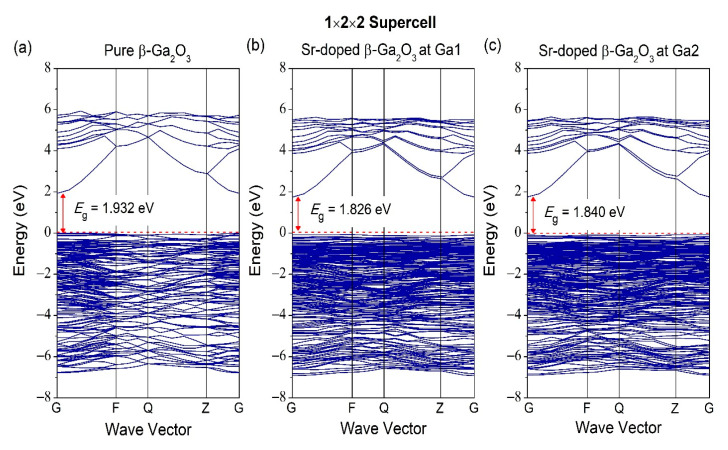
Band structure of (**a**) pure β-Ga_2_O_3_, (**b**) Sr-doped β-Ga_2_O_3_ at Ga1, and (**c**) Sr-doped β-Ga_2_O_3_ at Ga2 without scissor operator.

**Figure 8 micromachines-12-00348-f008:**
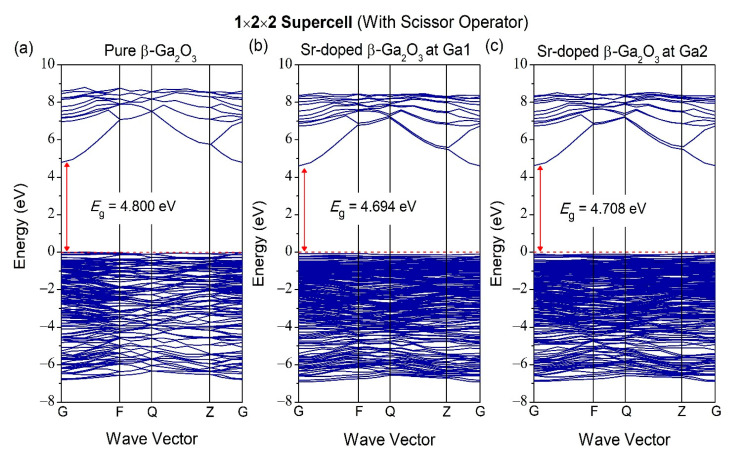
Band structure of (**a**) pure β-Ga_2_O_3_, (**b**) Sr-doped β-Ga_2_O_3_ at Ga1, and (**c**) Sr-doped β-Ga_2_O_3_ at Ga2 with scissor operator.

**Figure 9 micromachines-12-00348-f009:**
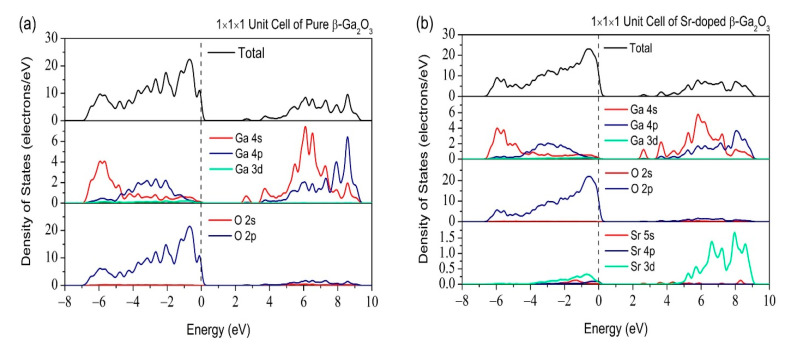
Total and partial density of states of (**a**) pure β-Ga_2_O_3_ and (**b**) Sr-doped β-Ga_2_O_3_ in 1 × 1 × 1 supercell.

**Figure 10 micromachines-12-00348-f010:**
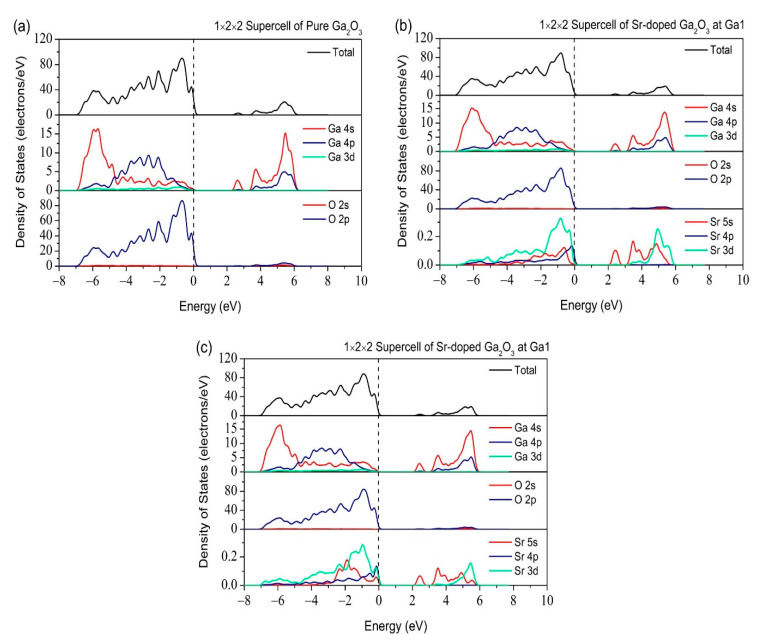
Total and Partial density of states of (**a**) pure β-Ga_2_O_3_, (**b**) Sr-doped β-Ga_2_O_3_ at Ga1, and (**c**) Sr-doped β-Ga_2_O_3_ at Ga2 in 1 × 2 × 2 supercell.

**Figure 11 micromachines-12-00348-f011:**
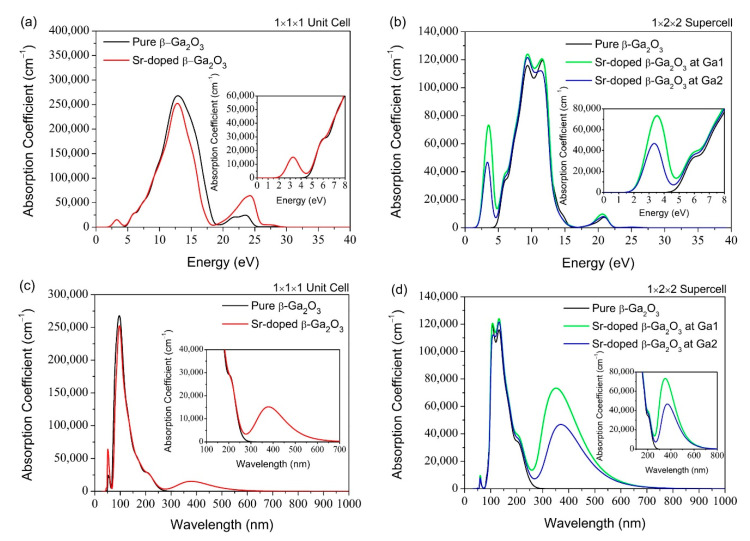
Absorption coefficient against (**a**) Energy (eV) for 1 × 1 × 1 pure and Sr-doped β-Ga_2_O_3_, (**b**) Energy (eV) for 1 × 2 × 2 supercell pure and Sr-doped β-Ga_2_O_3_, (**c**) Wavelength (nm) for 1 × 1 × 1 unit cell pure and Sr-doped β-Ga_2_O_3_, (**d**) Wavelength (nm) for 1 × 2 × 2 supercell for pure and Sr-doped β-Ga_2_O_3._

**Figure 12 micromachines-12-00348-f012:**
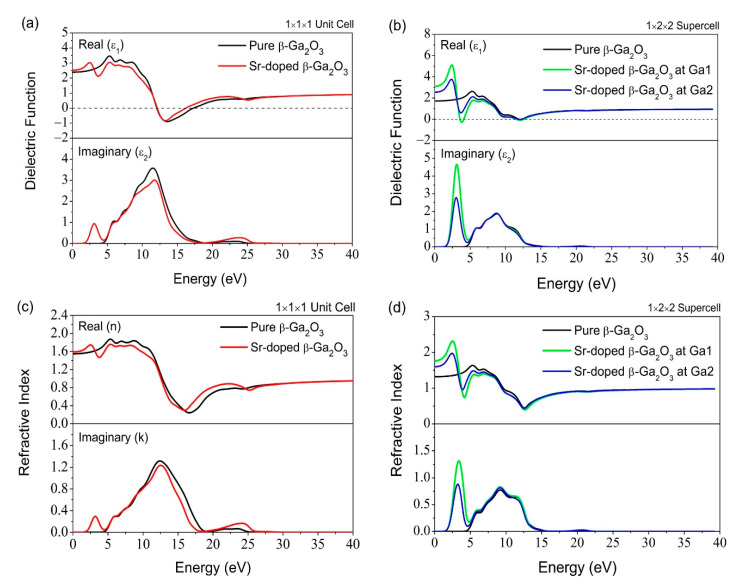
(**a**) Dielectric Function 1 × 1 × 1 unit cell, (**b**) Dielectric Function 1 × 2 × 2 supercell, (**c**) Refractive Index 1 × 1 × 1 unit cell, (**d**) Refractive Index 1 × 2 × 2 supercell.

**Figure 13 micromachines-12-00348-f013:**
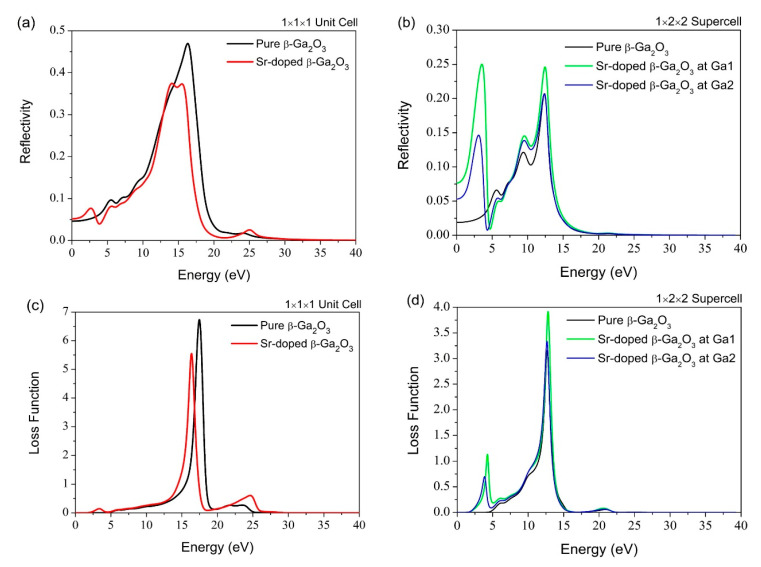
(**a**) Reflectivity of 1 × 1 × 1 unit cell, (**b**) Reflectivity of 1 × 2 × 2 supercell, (**c**) Loss Function of 1 × 1 × 1 unit cell, (**d**) Loss Function of 1 × 2 × 2 supercell.

**Table 1 micromachines-12-00348-t001:** Calculated lattice parameter (*a*, *b* and *c*), cell volume, and cell angle for pure and Sr-doped β-Ga_2_O_3_ crystal structure in 1 × 1 × 1 unit cell.

Structures	Parameters	GGA-PBE	Experiment [37]
β-Ga_2_O_3_	*a* (Å)	12.494 (+2.29%)	12.214
	*b* (Å)	3.096 (+1.94%)	3.037
	*c* (Å)	5.898 (+1.72%)	5.798
	Cell Volume (Å^3^)	221.647 (+6.14%)	208.835
	Cell Angle (°)	α = 90° β = 103.705° γ = 90°	α = 90° β = 103.830° γ = 90°
Sr-doped β-Ga_2_O_3_	*a* (Å)	12.506 (+0.10%)	-
	*b* (Å)	3.158 (+2.00%)	-
	*c* (Å)	5.794 (–1.76%)	-
	Cell Volume (Å^3^)	222.317 (+0.30%)	-
	Cell Angle (°)	α = 90°	-
		β = 103.701°	-
		γ = 90°	-

**Table 2 micromachines-12-00348-t002:** Calculated lattice parameter (*a*, *b*, and *c*), cell volume, and cell angle of pure and Sr-doped β-Ga_2_O_3_ crystal structure in 1 × 2 × 2 supercell at Ga1 and Ga2.

Parameters	Pure β-Ga_2_O_3_	Sr-Doped β-Ga_2_O_3_ at Ga1	Sr-Doped β-Ga_2_O_3_ at Ga2
*a* (Å)	12.497	12.545 (+0.38%)	12.599 (+0.82%)
*b* (Å)	6.187	6.218 (+0.50%)	6.233 (+0.74%)
*c* (Å)	11.806	11.937 (+1.11%)	11.857 (+0.43%)
Cell Volume (Å^3^)	886.770	903.865	904.761
Cell Angle (°)	α = 90° β = 103.723° γ = 90°	α = 90° β = 103.903° γ = 90°	α = 90° β = 103.667° γ = 90°

**Table 3 micromachines-12-00348-t003:** Calculated average bond length of pure β-Ga_2_O_3_ and Sr-doped β-Ga_2_O_3_ for 1 × 1 × 1 unit cell.

Bond Length	1 × 1 × 1 Unit Cell	
	Pure β-Ga_2_O_3_	Sr-Doped β-Ga_2_O_3_
Ga-O (Å)	1.964	1.965
O-O (Å)	2.861	2.864
Sr-O (Å)	-	2.426

**Table 4 micromachines-12-00348-t004:** Calculated average bond length of pure β-Ga_2_O_3_ and Sr-doped β-Ga_2_O_3_ for 1 × 2 × 2 unit cell.

Bond Length	1 × 2 × 2 Supercell		
	Pure β-Ga_2_O_3_	Sr-Doped β-Ga_2_O_3_ at Ga1	Sr-Doped β-Ga_2_O_3_ at Ga2
Ga-O (Å)	1.971	1.979	1.973
O-O (Å)	2.882	2.873	2.876
Sr-O (Å)	-	2.229	2.366

**Table 5 micromachines-12-00348-t005:** Indicator of population analysis for electron density.

Bonding Characteristics	*Ionic*	*Weak Ionic*	*Weak Covalent*	*Covalent*
**Bond Population**	0–0.32	0.32–0.50	0.50–0.75	0.75–1.0

**Table 6 micromachines-12-00348-t006:** Calculated optical properties without and with scissor operator (SO) of pure Ga_2_O_3_ and Sr-doped β-Ga_2_O_3._

	Pure Ga_2_O_3_		Sr-Doped β-Ga_2_O_3_	
	without SO	with SO	without SO	with SO
Absorption edge	400–700 nm	100–300 nm	400–700 nm	100–300 nm
Dielectric constant	3.05	2.39	5.37	2.53
Refractive index	1.75	1.55	2.33	1.59
Reflectivity peak	13.3 eV	16.3 eV	11.1 eV	15.5 eV
Loss function peak	14.3 eV	17.4 eV	13.3 eV	16.3 eV

**Table 7 micromachines-12-00348-t007:** Calculated optical properties without and with scissor operator (SO).

	Pure Ga_2_O_3_		Sr-Doped β-Ga_2_O_3_ at Ga1		Sr-Doped β-Ga_2_O_3_ at Ga2	
	without SO	with SO	without SO	with SO	without SO	with SO
Absorption edge	400–700 nm	100–300 nm	400–700 nm	100–300 nm	400–700 nm	100–300 nm
Dielectric constant	2.18	1.74	14.7	3.11	9.63	2.55
Refractive index	1.48	1.32	3.88	1.76	3.13	1.60
Reflectivity peak	9.46 eV	12.3 eV	9.46 eV	3.54 eV	9.33 eV	12.3 eV
Loss function peak	9.69 eV	12.7 eV	9.69 eV	12.8 eV	9.57 eV	12.5 eV

## Data Availability

No new data were created or analyzed in this study. Data sharing does not apply to this article.

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
