# Peer review of "First-Principles Studies for Electronic Structure and Optical Properties of Strontium Doped β-Ga2O3"

_micromachines, 2021, doi:10.3390/mi12040348_

Round 1
Reviewer 1 Report
file attached

Author Response
Dear Reviewer:
Thank you for your comments concerning our manuscript entitled "First-principles studies for electronic structure and optical properties of Strontium doped β-Ga2O3". Those comments are all valuable and very helpful for revising and improving our paper and the important guiding significance to our research. We have studied the comments carefully and have corrected them, hoping to meet with approval. All revised portions have been marked in red in the revised manuscript. The main corrections in the paper and responses to the reviewer are attached.

Reviewer 2 Report
The manuscript reports on the first-principles study of Sr-doped beta-Ga2O3. It follows multiple works devoted to this material using similar approaches. The originality consists in the used Sr dopant, which was not considered before. The results thus deserve to be published in Micromachines. Nevertheless, I identified several shortcomings lowering the benefits of the paper for interested researchers. Though some of them may be hardly repaired, authors should frankly address corresponding issues. The list of respective items follows.
1. The work is debased by chosen small cell consisting of 20 atoms only. For example, a supercell with as much as 120 atoms was used in [L. Dong et al., Comp. Mat. Sci. 156 (2019) 273]. The proximity and interaction of Sr doping atoms may affect the results. Authors should explain, why they used the small cell. Authors should also discuss the mutual interaction of Sr located in neighboring cells and estimate the effect to presented results. They should also assess the properties of a single Sr_Ga point defect in pure Ga2O3 lattice resulting from their computations.
2. The statements in lines 166-167 "When the concentration of Ga^3+ .... for semiconductors." is impenetrable. What does it mean the "increase of Ga^3+ concentration" in the material Ga2O3 where the Ga density is fixed? Similarly, the statement on the n-type behavior is not documented by the data. Also the next statements in lines 168 and 180 are confusing. Doping at a low density, either with donors or acceptors does not change the band gap. The seeming reduction of the band gap induced by Sr doping is due to the periodicity of Sr distribution in the supercell. The band gap does not change in reality in weakly doped materials and Sr forms an acceptor level near the valence band maximum.
3. For the comparison of the effect of Sr doping the band structures in Fig. 3 (a) and (b) should be plotted along the same trajectories through the Brillouin zone. Brillouin zone should be shown or coordinates of points defining the band structure profile Z,G,Y,A,B,D, ... should be defined.
4. The used approach amending big difference of the calculated band gap from experiment by the scissor operator should be substantiated. An error introduced by this approach to the calculation of respective quantities, Eqs. (1)-(5), should be estimated.
English needs revision. There are many trivial faults in grammar. For example in Abstract ".. analysis exhibit ..." or "... optical properties ... ... causes ...", line 37 "Other metal oxide structure ..."
There are also strange statements: line 49 "... can sustain large voltage applications for high voltage applications."
Author Response
Dear Reviewer:
Thank you for your comments concerning our manuscript entitled "First-principles studies for electronic structure and optical properties of Strontium doped β-Ga2O3". Those comments are all valuable and very helpful for revising and improving our paper and the important guiding significance to our research. We have studied the comments carefully and have corrected them, hoping to meet with approval. All revised portions have been marked in red in the revised manuscript. The main corrections in the paper and responses to the reviewers' comments are attached.

Round 2
Reviewer 1 Report
file attached

Round 3
Reviewer 1 Report
file attached

Author Response
Dear Reviewers,
I would like to thank the reviewers for their critical reading of our manuscript and their concrete criticism, which helped me revise and improve the manuscript. In this response letter, we respond to all of the criticisms and suggestion and the changes are written in red colour all in the manuscript. Our point-by-point responses to comments are detailed in the attached file.
We think that we have taken all suggestions by the referees into account in the revised version of the manuscript and would be happy if you would give it further consideration.
Best regards,
Mohd Ambri Mohamed
Associate Professor (Corresponding Author)

Round 4
Reviewer 1 Report
file attached
